# Factors influencing the use of multiple HIV prevention services among transport workers in a city in southwestern Uganda

**Benjamin Betunga[1], Phionah Atuhaire[1], Catherine Nakasiita[1], Christa Kanyamuneza[1], Proscovia Namiiro[1], Joseph Tugume[1], Matovu Hairat[1], Ahmed M. Sarki[2,3], Benedicto Mugabi[4], Birungi Lilian[5], Richard Mugisha[6], Edward Kumakech[7], John Baptist Asiimwe[2] \***

**1** Faculty of Nursing and Health Sciences, Bishop Stuart University, Mbarara, Uganda, **2** Aga Khan University, Uganda Campus, Kampala, Uganda, **3** Family and Youth Health Initiative (FAYOHI), Dutse, Jigawa State, Nigeria, **4** Baylor College of Medicine Children's Hospital, Kampala, Uganda, **5** College of Nursing, Sultan Qaboos University, Muscat, Oman, **6** Mbarara University of Science and Technology, Mbarara, Uganda, **7** Faculty of Nursing and Midwifery Lira University, Lira, Uganda

\* john.asiimwe@aku.edu

## Abstract

The use of multiple HIV prevention services has been found to decrease the risk of acquiring HIV when tailored to individuals at risk of HIV exposure, including transport workers. Therefore, we assessed the uptake of multiple HIV prevention services (≥2) and associated factors among transport workers in a city in Southwestern Uganda. This cross-sectional study comprised motorcycle taxi riders, motor vehicle and truck drivers, aged 18 to 55 years who were selected and responded to an interviewer-administered questionnaire, between November 2021 and February 2022. Data was analyzed using descriptive statistical and modified Poisson regression analyses. Out of 420 participants, 97.6% were male, with a median age of 28 years and the majority were aged <34 years (84.6%). Overall, less than half (45.3%) of the participants had used multiple (≥2) HIV prevention services within a one-year period. Many participants had used condoms (32.2%) followed by voluntary HIV counseling and testing (27.1%), and safe male circumcision (17.3%). Most participants who tested for HIV had ever used condoms (16.2%), followed by those who received safe male circumcision and had ever used condoms (15%), and those who tested for HIV and had started on antiretroviral therapy (ART) (9.1%). In the adjusted model, factors that were significantly associated with the use of multiple HIV prevention services included religion (aPR = 1.25, 95% CI = 1.05–1.49), the number of concurrent sex partners (aPR = 1.33, 95% CI = 1.10–1.61), prior HIV testing and awareness of HIV serostatus (aPR = 0.55, 95% CI = 0.43–0.70), awareness of HIV prevention services (aPR = 2.49, 95% CI = 1.16–5.38), and financial payment to access HIV services (aPR = 2.27, 95% CI = 1.47–3.49). In conclusion, the uptake of multiple HIV prevention services among transport workers remains suboptimal. Additionally, individual behavioral factors influence the use of multiple HIV services compared with other factors. Therefore, differentiated strategies are needed to increase the utilization of HIV prevention services among transport workers.

**Data Availability Statement:** All relevant data are within the paper and its Supporting Information file.

**Funding:** The research reported in this publication was partly supported by the Fogarty International Center (U.S. Department of state's Office of the US Global Aids Coordinator and Health Diplomacy (S/GAC) and the president's Emergency Plan for AIDs Relief (PEPFAR) of the national institutes of Health under the award number R25TW011210 to BB. The content is solely the responsibility of the authors and does not necessarily represent the official views of the National Institutes of Health. The funder had no role in the study design, data collection, analysis, decision to publish, or preparation of the manuscript. No relevant grant or award recipients are specifically associated with the funding received for this study.

**Competing interests:** The authors have declared that no competing interests exist.

## Introduction

Over the last three decades, there has been tremendous progress in curbing the HIV epidemic in Uganda. HIV prevalence has declined from 18% in the 1990s to 6.2% in 2016 and 5.4% in 2020 among 15–49-year-olds [1–3]. Notwithstanding the high mortality rates among patients with acquired immune deficiency syndrome (AIDs), the decline in the prevalence of HIV is "partly" attributed to the use of HIV prevention services such as condom use, and antiretroviral based prevention services, like in many other sub-Saharan countries [4].

One of the current drivers of the HIV epidemic that threatens to undermine the success story is the high incidence of HIV among high-risk groups or key populations [2, 5]. Transport workers which include motorcycle taxi riders, truck drivers, and motor vehicle taxi drivers as defined for this study are among the high-risk groups for acquiring HIV [6–11]. HIV prevalence among transport workers varies across countries. It ranged from 9.2 to 13.9% among truck drivers in Cambodia [12]. In some sub-Saharan Africa countries, the prevalence exceeds the national average for HIV prevalence [11]. As an example, the prevalence of HIV among truck drivers in South Africa and motorcycle taxi riders in Mbarara city in south-western Uganda were 26% and 9.9%, much higher than the national averages of 12% and 5.4%, respectively [13–15].

The high HIV prevalence among transport workers, particularly long-distance truck drivers' is attributable to engaging in transactional sex with sex workers along their transit routes, risky sexual behaviors such as having multiple concurrent sex partners and alcoholism, lack of access to and or poor utilization of HIV prevention services, low education, lack of knowledge, and awareness of HIV prevention methods among other factors [10, 16, 17]. In fact, transport workers, especially truck drivers make up to 70% of the clients of sex workers, and about 26% of the new HIV infections in sub-Saharan Africa in 2021 [15, 18, 19].

Given that HIV transmission is driven by multiple biomedical, behavioral, and structural factors, the use of a single HIV prevention method by high-risk populations is unlikely to alter the HIV epidemic trajectory, putting the attainment of the 2025 goal of less than 370, 000 annual new infections worldwide in jeopardy [5, 18, 20, 21]. To maximize population-level effects and potentially bring the HIV epidemic under control, countries (Uganda inclusive) have undertaken a policy to provide HIV prevention services in series at health facilities and during community health outreaches to enable multiple uptake of these services [2, 20, 22]. The outcomes of this policy shift has been impressive so far. For instance, using multiple HIV prevention services was found to decrease the risk of acquiring HIV when tailored to individuals at high risk of HIV exposure or key populations. In Kenya, providing multiple HIV prevention services to key populations reduced the incidence of new HIV infections by 44%, whereas in Thailand, Cambodia, and Vietnam, new HIV infections reduced by over 60% between 2010 and 2020 [18, 21].

Mechanisms behind the beneficial effect of the use of multiple HIV prevention services on HIV infections are well known. The prompt use of multiple HIV prevention services such as voluntary HIV counselling and testing (VCT) followed by antiretroviral therapy (ART) preferably within a year of HIV diagnosis is critical in preventing delays in treatment initiation, reducing missed opportunities for treatment, preventing HIV progression and ultimately reducing the HIV transmission risk [23]. In fact, the prompt use of VCT and antiretroviral drugs (ARVs) for the elimination of mother-to-child transmission of HIV (EMTCT) series of HIV services is critical in the prevention of mother-to-child transmission of HIV (MTCT) [24]. Whereas the use of the other series of HIV prevention services such as VCT and condom use or Preexposure prophylaxis (PrEP) are critical in preventing new HIV infections among HIV sero-discordant partners [25].

Given the aforementioned significance, the WHO and ministries of health across sub–Saharan Africa including Uganda have instituted policies that enable multiple uses of HIV prevention services such as "test and treat" for HIV test-positive cases, "test, and condoms" or "test and PrEP" for HIV sero-discordant couples, "test, and PEP (post-exposure prophylaxis)" for occupational or accidental or forced exposure cases. There is also the policy of "test and ART" for EMTCT for pregnant and breast-feeding women living with HIV [2, 20].

Despite the availability of policies and a multitude of HIV prevention services at various health service delivery points in Uganda, the uptake of multiple HIV prevention services and factors associated with its use, especially among transport workers, remains poorly documented. A few previous studies such as the one conducted among men who have sex with men (MSM) in Brazil, found low (2.8%) use of behavioral and biomedical HIV prevention services [26]. Additionally, previous research findings from South Africa indicated suboptimal use of HIV prevention services by transport workers, despite interventions by various stakeholders to increase transport workers' access to the HIV services [7, 8, 11].

Individual factors such as sociodemographic characteristics, knowledge, and awareness about HIV prevention, perceived risk of HIV infection, and HIV stigma among others, were reported to predict the use of single HIV prevention services. In agreement, good knowledge about HIV transmission, perceived high risk of acquiring HIV infection, awareness of HIV status, and having a high level of education among others, were found to be associated with higher likelihood of using either condoms or HIV testing services among truck drivers and motorcycle taxi riders in multi-country studies which included Togo, Nigeria, and Uganda, respectively [17, 27, 28]. In this study, we hypothesized these factors as determinants of the uptake of multiple HIV prevention services among transport workers through similar mechanisms.

Therefore, we set out to investigate the influence of individual factors on the utilization of multiple HIV prevention services among transport workers in southwestern Uganda. The findings of this study will inform tailor-made interventions aimed at increasing access to and utilization of HIV prevention services among transport workers in Uganda and other similar sub-Saharan Africa settings.

## Materials and methods

### Study design and setting

This study used a cross-sectional design. The study setting was the four slum areas of Mbarara city in southwestern Uganda where the HIV prevalence was higher than the Uganda national average. The slum areas (also known as hot spots) of Mbarara city included Kakoba, Kiyanja, Kizungu, and Ruti. The areas have many sex workers, motorcycle taxi riders, motor vehicle taxi drivers (who are residents of the area and its neighborhood), and stopover stations for truck drivers on transit. According to the 2020 estimates, Mbarara district has the third highest level of HIV prevalence (13.1%) in Uganda when compared to the national average of 5.4% [2, 18]. Among a subset of transport workers in this case motorcycle riders (also known as boda-bodas) in a Mbarara city, a previous study reported a 9.9% prevalence of HIV infections which was also above the national average of 5.4% [13].

There are several health facilities in and around the four slum areas of Mbarara city that provide all the recommended HIV prevention services including Kakoba Health Center, Nyamityobora health center, Mbarara City Hospital, and Mbarara Regional Referral Hospital. Besides the health facilities, there are other non-governmental organizations such as The AIDS Support Organization (TASO) that also provide HIV prevention services at mobile community outreach events to people living in Mbarara city's slum areas. Mbarara city is located in

southwestern Uganda at a distance of about 280 kilometers from Kampala the capital city of Uganda.

## Sample, sample size, and sampling method

We included in the study male and female motorcycle taxi riders, and motor vehicle taxi and truck drivers aged 18 and 55 years, who were residents and have operated their transport trade within Mbarara city for at least one year preceding the study. We excluded transport workers who were feeling sick or unwell and thus unable to withstand the study procedures. The initial study sample size of 385 transport workers was obtained using the Cochrane (1963) formula [29]. For calculating the sample size, the proportion of transport workers utilizing multiple HIV prevention services *(p)* was assumed to be 50% ($p = 0.5$), the Z score from standard normal curve corresponding with the 95% confidence interval, and the allowable error were set at 1.96 and 0.05 respectively. We adjusted the initial sample size of 385 participants for 10% non-response rate to give the desired sample size of 424 participants. From the register of transport workers obtained from their leaders, a complete register was established. We performed simple random sampling to select the participants for the study. Out of the 424 participants sampled and approached for informed consent, 422 participants provided informed consent and participated in the study (a 99.5% response rate). Only two participants declined to provide informed consent and participate in the study because of their busy work schedules.

## Data collection procedure

The data were collected between November 2021 and February 2022. We obtained the transport workers' registers with their contact details from their leaders. Simple random sampling (the lottery method with replacement) was used to select eligible participants. The leaders of the respective transport workers then contacted the selected participants by phone calls and introduced the team of researchers to them. This was followed by appointments to meet the study participants. When a participant declined to participate in the study, two failed follow up attempts were made before a replacement was sampled from the register to participate in the study. Prior to the commencement of data collection, the principal investigator trained research assistants (who were Bachelor of Nursing professionals) about the study protocol including the study tools. The research assistants participated in pretesting of the questionnaires on 10 participants which included motorcycle taxi riders, motor vehicle taxi, and truck drivers at another slum area known as Ruharo trading center within Mbarara city. Adjustments to the study tools were made to assure validity before their use in the main data collection.

At the respective stopovers or workplaces of transport workers, the research assistants identified the selected participants, introduced, and informed them about the study, and requested their participation in the study. Motorcycle riders who accepted to participate in study would then be moved aside from the other members of the group for the informed consent, and data collection to assure privacy and confidentiality. Tax drivers and truck drivers were interviewed in their vehicles when they did not have customers to assure privacy and confidentiality. All data were collected in the local languages (Runyakole-Rukiga) or English. Each data collection procedure took about 10–15 minutes. After each data collection session, the research assistants checked the questionnaire for completeness before leaving the participant and the field site. The two research assistants of nursing educational background who worked in a pair collected all the data.

## Measurement of the use of multiple HIV prevention services

Using multiple HIV prevention services was the primary outcome of interest for the study. Participants were asked to mention the HIV prevention services they had ever used within the previous 12 months. Using multiple HIV prevention services was indicated when a participant used two or more of the recommended HIV prevention services within a 12 months' period. The HIV prevention services comprised evidence-informed high-impact biomedical and behavioral HIV prevention services and methods such as ARVs for the EMTCT or ART, condoms, VCT, health education on HIV, safe male circumcision (SMC), PrEP and PEP. Participants were also asked to mention the source where the services were obtained from and how often they used the HIV services within the 12 months' period.

## Measurements of factors of interest

We categorized the factors which were potential predictor variables into three namely the participant's socio-demographic characteristics, knowledge, awareness and perceptions (KAP) and behavioral factors among the study participants:

**1. Sociodemographic characteristics.** The participants were asked to state their status on nine items which were considered as measures of the sociodemographic characteristics of the participants namely age, gender, marital status, education level, income level, occupation, length of stay in the study area, distance to a nearby health facility, and religion.

**2. Knowledge, awareness and perception (KAP) factors among the participants'.** *a). Awareness of HIV prevention services.* Participants were asked to mention any biomedical or behavioral HIV prevention services they were aware of and their source of information. Awareness was categorized into awareness of a single (1) service and awareness of multiple HIV prevention services (2+) as measures of awareness.

*b) Knowledge of HIV prevention.* Participant's knowledge about HIV prevention was assessed using a validated questionnaire adapted from a previous study conducted in South Africa and Cambodia [30, 31]. The questionnaire has twelve knowledge items about HIV prevention with answer options of "Yes", "No", or "Don't Know". A score of 1 was awarded to participants who correctly answered a question, and 0 for incorrect or don't know responses. The maximum attainable score was twelve (12) which was further grouped into poor knowledge level (0–6 score), and good knowledge level (7–12 score).

*c) Perceived risk of HIV infection.* The perceived risk of HIV infection was measured using a validated perceived risk of HIV infection questionnaire adapted from Napper et al., [32]. The questionnaire measures the cognitive and affective aspects of the perceived risk of HIV infection, namely, how an individual conceptualizes the risk of HIV infection, feels about it, and how often they think about it. The tool has ten (10) risk statements with strongly agree to strongly disagree measured on a 5-point likert scale. Participants were asked to provide their level of agreement with each of the statements regarding the perceived risk of contracting HIV infection. The maximum attainable score was 43 points. The scores were further grouped into the low perceived risk of HIV infection (0–22.5) and high perceived risk of HIV infection (22.6–43.0).

*d) Perceived HIV stigma.* Perceived HIV stigma was measured using a validated 11-itemed questionnaire which was adapted from the questionnaire used to assess perceived personal and community HIV stigma in South Africa [33]. On a 5-point Likert scale (strongly agree to strongly disagree), participants were asked to provide their level of agreement with the 11 statements measuring stigma. The maximum attainable score was forty-four (44) which was further grouped into low perceived HIV stigma (0–22), and high perceived HIV stigma (23–44).

**Table 1. Internal consistency and measurement properties of items.**

| Subscales | Items | MIIC | Cronbach's α | NCITC>0.3 |
|---|---|---|---|---|
| Knowledge of HIV prevention | 12 | 0.162 | 0.707 | 7 |
| Perceived risk of HIV* | 8 | 0.497 | 0.88* | 8 |
| Perceived HIV Stigma | 11 | 0.483 | 0.91 | 11 |

**Note:** The coefficient

* in the perceived risk of HIV subscale means two items were eliminated from the scale to compute the coefficient as recommended by the authors of the tool (Napper et al. [32]).

**3. Behavioral factors.** The participants were asked to state their status regarding the three (3) items which were considered as measures of behaviors likely to influence the access and or utilization of HIV prevention services. These behaviors included sexual behavior in particular the number of concurrent sex partners, HIV self-care behavior in particular prior HIV testing (as well as knowing own HIV status) and willingness to pay or having ever paid to access or obtain HIV prevention services.

## Reliability analyses

Reliability analyses were conducted for knowledge of HIV prevention, perceived risk of HIV infection, and HIV stigma subscales. Before reliability analyses, all negatively worded items plus their respective scores within the perceived risk of HIV infection and perceived HIV stigma subscales were reversed. Internal consistency of the subscales was evaluated using the Cronbach alpha (α) coefficient. The Cronbach alpha (α) coefficient values of 0.7 or higher were considered acceptable [34], see Table 1). Overall, the Cronbach alpha coefficient for the knowledge of HIV prevention, perceived risk of HIV infection, and perceived HIV stigma subscales were 0.707, 0.88, and 0.91, respectively, demonstrated all the three subscales were reliable at measuring the construct they were meant to measure. Notably, both the mean inter-item correlation (MIIC) and the number of items with corrected item-total correlation (NCITC, >0.3) were within acceptable limits.

## Data analysis

The data were entered and cleaned in EpiData 3.1 software and exported to the Statistical Package for Social Sciences (SPSS) version 20 for analysis (S1 Data). The primary outcome variable (with response score options of 1 or 0) was created for the purpose of conducting a modified Poisson regression analysis. Descriptive analyses were conducted for categorical and continuous variables. Categorical variables are reported as frequency counts and percentages whereas continuous variables were summarized as mean (when normally distributed) and median (when not normally distributed). Chi-square statistics were used to establish the relationship between categorical factors and the use of multiple HIV prevention services. Variables found statistically significantly associated with the use of multiple HIV prevention services (p<0.05) were entered into the modified Poisson regression models including demographic characteristics, KAP factors, and behavioral factors to obtain their predictive and explanatory effects. The level of significance and adjusted prevalence ratios (aPRs) were reported at 95% confidence intervals.

## Ethical consideration

The Mbarara University of Science and Technology research ethics committee (MUREC 1/7/ #15/12-20) approved the study. Administrative permission to conduct the research was

obtained from the district health officer of the Mbarara district and the leaders of the transport workers. Informed consent was obtained from all study participants after introducing the study to them using the informed consent form. Using either a thumbprint (illiterate) or a signature confirmed consent. For confidentiality, participants were interviewed away from the other transport workers in private spaces. The consenting and interview processes were conducted in the local languages (Runyakole-Rukiga) or English. Interviews were stopped, and participants were dropped and replaced if they received a customer for their transportation trade during the study and this happened with 26 participants. Research assistants and participants wore masks and disinfected their hands with alcohol gel to avoid cross-infection with COVID-19. We paid time compensation with an amount equivalent to USD 2 to every participant upon completion of the data collection session. Copies of the completed questionnaires were kept in a lockable cabinet in the first author's office whereas the electronic databases were password protected and only accessible to the research team.

## Results

### Demographic characteristics of the study participants

The demographic characteristics of the study participants are summarized in Table 2. Out of the 420 participants interviewed (99.5% response rate), the majority were male (97.6%) and middle-aged adults aged below 34 years, with a median age of 28 years (54.4%). About half of the study participants had lived, temporarily stayed, or worked in Mbarara city for over 5 years (51.0%) with a median of 6 years, in the same city. The majority were single (41.2%), Christians by religion (78.6%), and had obtained a primary (38.5%) or ordinary-level certificate of education (39.5%). Most participants were motorcycle taxi riders (boda-boda riders, 69.7%), followed by motor vehicle taxi and truck drivers (25.9%). Most earned a monthly income of 400,001–500,000/ = (~111 to137 USD, 59.5%). Many of the study participants knew their current HIV serostatus (54.1%); most were seronegative (43.9%). However, a large proportion of participants had multiple concurrent sex partners (52.4%), with a median of 2 partners per participant. Last, most participants lived, stayed, or worked within 1–3 kilometres (km) of the nearby health facility (86.6%), with a median of 2 km.

### The use of multiple HIV prevention services

When participants were asked to disclose the number of HIV prevention services they had ever used within the past 12 months, results indicate that less than half of the participants (45.3%), had ever used two or more services (Table 3). Approximately one-third of the participants had used condoms (32.2%) followed by voluntary counselling and testing (27.1%), and safe male circumcision (17.3%).

Regarding multiple use of HIV prevention services, the findings show that a relatively low proportion of the participants had tested for HIV and then used condoms (16.2%), followed by those who had received safe male circumcision and then used condoms (15%), and those who had tested for HIV and then began on ART (9.1%). Most of the HIV prevention services were obtained from the public health facilities and not for profit civil society organizations that support patients with HIV (83.3%) and were free (82.8%).

### Factors associated with the use of multiple HIV prevention services

The results of the bivariate analyses shown in Table 4 indicate that participant's sociodemographic characteristics namely age, education level, marital status, religion and years of residence (or work) in the study area were significantly associated with the utilization of multiple

**Table 2. Demographic characteristic of the participants.**

| Variable | n (%) |
|---|---|
| **Gender** | |
| Male | 400 (97.6) |
| Female | 10 (2.5) |
| **Age in years** | **28 (18–48)*** |
| 18–24 yrs. | 134 (30.2) |
| 25–34 yrs. | 223 (54.4) |
| ≥ 35 yrs. | 63 (15.4) |
| **Length of stay in the area (yrs.)** | **6 (1–28)*** |
| <5years | 201 (49.0) |
| ≥ 5year | 209 (51.0) |
| **Marital status** | |
| Single | 169 (41.2) |
| Married | 154 (37.6) |
| Cohabiting | 22 (5.4) |
| Separated/ divorced/ widowed | 65 (15.9) |
| **Religion** | |
| Anglican | 131 (32.0) |
| Catholic | 118 (28.8) |
| Pentecost | 73 (17.8) |
| Moslem | 84 (20.5) |
| Others | 4 (1.0) |
| **Education level** | |
| None | 23 (5.6) |
| Primary | 158 (38.5) |
| Ordinary level | 162 (39.5) |
| Advanced level | 56 (13.7) |
| Tertiary | 11 (2.7) |
| **Occupation** | |
| Motorcycle rider (Boda Boda) | 286 (69.7) |
| Motor vehicle driver/conductor | 106 (25.9) |
| Truck driver/conductor | 18 (4.4) |
| **Monthly income** | |
| 82–110 USD | 48 (11.7) |
| 111-137USD | 244 (59.5) |
| >137 USD | 118 (28.8) |
| **Serostatus (self-report)** | |
| Positive | 42 (10.2) |
| Negative | 180 (43.9) |
| Unknown | 188 (45.9) |
| **Sex partners** | **2 (0–11)*** |
| Single | 195 (47.6) |
| Multiple | 215 (52.4) |
| **Distance to health facility/HIV care service provider** | **2 (1–10)*** |
| 1-3km | 355 (86.6) |
| +3km | 55 (13.4) |

Key: ***Median (Minimum-Maximum)**

**Table 3. Use of HIV prevention services among transport workers in Mbarara city, southwestern Uganda (N = 420).**

| Variables | n (%) |
|---|---|
| **Number of Used HIV prevention services** | |
| 1 or None | 223 (54.7) |
| ≥ 2 services | 185 (45.3) |
| **Use of a single HIV prevention service.** | |
| VCT | 171 (27.1) |
| Condom Use | 203 (32.2) |
| SMC | 109 (17.3) |
| PREP | 10 (1.6) |
| EMTCT/ART | 66 (10.5) |
| Others | 43 (6.8) |
| **Common combinations used** | |
| VCT + ART/EMTCT | 38 (9.1) |
| VCT + PrEP | 06 (1.4) |
| VCT + condoms | 68 (16.2) |
| VCT + SMC | 36 (8.6) |
| SMC+ condoms | 63 (15) |
| **Sources of HIV prevention services** | |
| Government | 197 (56.8) |
| CSO and others | 92 (26.5) |
| Private health facility | 58 (16.7) |
| **Pay for services** | |
| Yes | 58 (16.9) |
| No | 280 (82.8) |

*Note*: VCT: voluntary testing and counselling; SMC: safe male circumcision; PrEP: pre-exposure prophylaxis; EMTCT: elimination of mother-to-child transmission of HIV; ART: Anti-retroviral therapy; CSO: civil society organizations.

HIV prevention services. Similarly, the results indicate that awareness, knowledge, perceptions and behavioral factors were significantly associated with the utilization of multiple HIV prevention services.

The results of the univariate Poisson regression are also shown in Table 4. Participants who were older (≥25years) compared with the younger participants (<25years); participants who subscribed to Muslim or other religious faith compared with those who belonged to the Christian faith, were 1.376, and 1.381 more likely to use multiple HIV prevention services, respectively. In addition, participants who had worked for a longer period (> 5 years) around the city compared with those who had worked for a short period (< 5 years), and participants who were aware of two or more HIV prevention services compared with those who were aware of one, and participants who did not pay to access HIV prevention services compared with those who paid, were 1.283, 3.234, and 2.515 times more likely to use multiple HIV prevention services, respectively.

Contrariwise, participants who were not aware of their HIV status compared to those who knew it and participants who had perceived high level of HIV stigma compared with a low perceived HIV stigma were 0.512 and 0.808 times less likely to use multiple HIV prevention services, respectively. The other variables in Table 4 were not statistically significantly associated with the use of multiple HIV prevention services in univariate Poisson regression analysis.

**Table 4. Factors associated with the use of multiple HIV prevention services.**

| Variable | Use of multiple HIV Prevention services | | Pearson chi-square | P-value | Unadjusted prevalence ratios (uPRs) |
|---|---|---|---|---|---|
| | 2 and + (n, %) | 1 or None n (%) | Value (d.f) | Asymp. sig. (2-sided) | uPRs (95% CI) |
| **Overall** | 185 (45.3) | 223 (54.7) | | | |
| **Gender** | | | 0.90 (1) | 0.765 | |
| Male | 180 (44.1) | 218 (53.4) | | | 1.00 |
| Female | 5 (1.2) | 5 (1.2) | | | 1.106 (0.660–1.852) |
| **Occupation** | | | 0.999 (2) | 0.607 | |
| Motorcycle taxi rider (Boda Boda) | 125 (30.6) | 160 (39.2) | | | 1.00 |
| Motor vehicle Driver/conductor | 52 (12.7) | 53 (13.0) | | 0.319 | 1.184 (0.961–1.459) |
| Truck Driver/conductor | 8 (2.0) | 10 (2.5) | | 0.961 | 1.191 (0.762–1.862) |
| **Age (years)** | | | 15.349 (2) | 0.000* | |
| 18–24 yrs | 38 (9.3) | 85 (20.8) | | | 1.00 |
| 25–34 yrs | 112 (27.5) | 110 (27.0) | | 0.001* | 1.376 (1.051–1.803) |
| ≥35 yrs | 35 (8.6) | 28 (6.9) | | 0.001* | 1.454 (1.059–1.998) |
| **Number of Sex partners** | | | 3.938 (1) | 0.047* | |
| Single or none | 78 (19.1) | 116 (28.4) | | | 1.00 |
| Multiple (2 and +) | 107 (26.2) | 107 (26.2) | | | 1.165 (0.954–1.422) |
| **Marital status** | | | 5.997 (1) | 0.014* | |
| Not in relationship | 93 (22.8) | 139 (4.1) | | | 1.00 |
| In relationship | 92 (22.5) | 84 (20.6) | | | 1.155 (0.951–1.403) |
| **Religion** | | | 10.030 (1) | 0.002* | |
| Christians | 132 (32.4) | 188 (46.1) | | | 1.00 |
| Moslem and Others | 53 (13.0) | 35 (6.1) | | | 1.381 (1.139–1.675) |
| **Income level** | | | 0.184 (1) | 0.688 | |
| ≤500,000/ = (below average ~USD 137) | 130 (31.9) | 161 (39.5) | | | 1.00 |
| ≥500,000/ = (~USD 137) | 55 (13.5) | 62 (15.2) | | | 1.140 (0.928–1.400) |
| **Education level** | | | 5.414 (1) | 0.020* | |
| ≤Primary | 70 (17.2) | 110 (27.0) | | | 1.00 |
| ≥Secondary | 115 (28.2) | 113 (27.7) | | | 1.057 (0.863–1.294) |
| **HIV status** | | | 58.601 (1) | 0.000* | |
| Known | 139 (34.1) | 83 (20.3) | | | 1.00 |
| Unknown | 46 (11.3) | 140 (34.3) | | | 0.512 (0.396–0.660) |
| **Distance to health facility** | | | 1.315(1) | 0.251 | |
| 1-3km | 164 (40.2) | 189 (46.3) | | | 1.00 |
| >3km | 21 (5.1) | 34 (8.3) | | | 0.937 (0.673–1.303) |
| **Years of work in the area** | | | 4.519 (1) | 0.034* | |
| <5years | 80 (19.6) | 120 (29.4) | | | 1.00 |
| >5year | 105 (25.7) | 103 (25.2) | | | 1.283 (1.051–1.566) |
| **HIV Services Awareness** | | | 26.776 (1) | 0.000* | |
| 1 service | 5 (1.2) | 43 (10.5) | | | 1.00 |
| 2+ services | 180 (44.1) | 180 (44.1) | | | 3.234 (1.453–7.197) |
| **Knowledge about HIV prevention** | | | 10.028 (1) | 0.002* | |
| Poor | 42 (10.3) | 83 (20.3) | | | 1.00 |
| Good | 143 (35.0) | 140 (34.3) | | | 1.247 (0.972–1.599) |
| **Perceived risk of HIV** | | | 21.290 (1) | 0.000* | |
| Low | 25 (6.1) | 74 (18.1) | | | 1.00 |
| High | 160 (39.2) | 149 (36.5) | | | 1.324 (0.962–1.822) |

*(Continued)*

**Table 4.** (Continued)

| | Use of multiple HIV Prevention services | | Pearson chi-square | P-value | Unadjusted prevalence ratios (*u*PRs) |
|---|---|---|---|---|---|
| **Perceived stigma** | | | 11.733 (1) | 0.001* | |
| Low | 87 (21.3) | 68 (16.7) | | | 1.00 |
| High | 98 (24.0) | 155 (38.0) | | | 0.808 (0.666–0.979) |
| **Pay for HIV services** | | | 23.050 (1) | 0.000* | |
| Yes | 43 (12.7) | 15 (4.4) | | | 1 |
| NO | 111 (32.8) | 169 (50) | | | 2.515 (1.579–4.008) |

*Note*: D.f is degree of freedom, unadjusted prevalence ratios (*u*PRs)

### Factors that independently predict the use of multiple HIV prevention services

The factors that were significantly associated with the use of multiple HIV services (p<0.1) from the bivariate analyses were entered into three separate multivariate Poisson regression models (Table 5). The first model comprised sociodemographic factors such as age, education, religion, marital status, and length of stay in the area. In this model, results showed that age (*p* = 0.022), and religion (*p* = 0.01) independently predicted the use of multiple HIV prevention services. Participants aged between 24–25 years were 1.452 times more likely to use multiple HIV prevention services than those aged 18–24 years after controlling for other sociodemographic factors. In addition, participants who were Muslim or other faith were also 1.328 more likely to use multiple HIV prevention services than those who subscribed to the Christian faith after controlling for other sociodemographic factors.

In model 2, knowledge of HIV prevention, perceived risk of HIV, perceived HIV stigma, and awareness of HIV prevention services were then added to model 1. The results from model 2 multivariate regression model in Table 5 indicate that religion (*p* = 0.021), perceived risk of HIV (*p* = 0.005), and awareness of HIV services (*p* = 0.001) independently predicted the use of multiple HIV prevention services. Whereas age (*p* = 0.101) became statistically non-significant. Again, participants who subscribed to the Muslim or other faith were 1.281 times more likely to use multiple HIV prevention services than those who subscribed to the Christian faith. Participants who perceived the high risk of acquiring HIV were almost twice more likely to use multiple HIV prevention services than those who perceived low risk of HIV. Participants who were aware of two or more HIV prevention services were almost four times more likely to use multiple HIV prevention services than those who knew about only one HIV prevention service.

In model 3, behavioral factors in this case the number of concurrent sex partners, HIV testing, or awareness of HIV status, and paying for HIV prevention services were added into model 2. In model 3, religion (*p* = 0.011), awareness of HIV status (*p*<0.001), awareness of HIV prevention services (*p* = 0.019), paying for HIV prevention services (*p*<0.001), and the number of concurrent sexual partners (*p* = 0.003) independently predicted the use of multiple HIV prevention services. In this model 3, participants who subscribed to the Muslim or other faith were 1.253 more likely to use multiple HIV prevention services than those who subscribed to the Christian faith. Participants who had two or more sex partners were 1.330 times more likely to use multiple HIV prevention services than those with single or no sex partners. Additionally, Participants who were aware of two or more HIV prevention services were 2.498 times more likely to use multiple HIV prevention services than those who knew only one HIV prevention service. Participants who never paid to access the HIV prevention services were 2.267 times more likely to use HIV prevention services than those who paid to access the HIV

**Table 5. Prevalence ratios (and 95%confidence interval) from modified Poisson regression analysis of individual factors that predict the use of multiple HIV prevention services.**

| Variable | *a*PRs Exp (B) (95% CI) | | |
|---|---|---|---|
| | **Model 1** | **Model 2** | **Model 3** |
| **Age (years)** | | | |
| 18–24 yrs (Ref.) | 1.00 | 1.00 | 1.00 |
| 25–34 yrs | 1.452 (1.055–1.999)* | 1.285 (0.952–1.734) | 1.171 (0.906–1.512) |
| >35 yrs | 1.408 (0.932–2.128) | 1.308 (0.892–1.918) | 1.130 (0.796–1.603) |
| **Length of stay (years)** | | | |
| 5yrs &< (Ref.) | 1.00 | 1.00 | 1.00 |
| > 5 yrs | 1.179 (0.937–1.483) | 1.161 (0.934–1.443) | 1.087 (0.895–1.320) |
| **Religion** | | | |
| Christian (Ref) | 1.00 | 1.00 | 1.00 |
| Muslim & others | 1.328 (1.071–1.648)* | 1.281 (1.039–1.580) * | 1.253(1.053–1.491)* |
| **Education** | | | |
| Primary & none | 1.00 | 1.00 | 1.00 |
| Secondary & above | 1.135 (0.900–1.432) | 0.936 (0.748–1.171) | 0.907 (0.750–1.096) |
| **Marital status** | | | |
| Not in relationship | 1.00 | 1.00 | 1.00 |
| In relationship | 1.069 (0.844–1.354) | 0.975 (0.783–1.214) | 0.909 (0.746–1.108) |
| **Awareness of HIV services** | | | |
| Services | - | 1.00 | 1.00 |
| 2 services and above | - | 3.926 (1.729–8.918)* | 2.498 (1.160–5.376)* |
| **Knowledge about HIV** | - | | |
| Poor | - | 1.00 | 1.00 |
| Good | - | 1.121 (0.841–1.495) | 1.070 (0.823–1.391) |
| **Perceived risk of HIV** | - | | |
| Low (Ref) | - | 1.00 | 1.00 |
| High | - | 1.720 (1.176–2.515)* | 0.999 (0.733–1.361) |
| **Perceived stigma** | - | | |
| Low (Ref.) | - | 1.00 | 1.00 |
| High | - | 0.842(0.680–1.042) | 0.876 (0.726–1.057) |
| **Number of sex partners** | - | | |
| Single or none (Ref) | - | - | 1.00 |
| Multiple (2 and +) | - | - | 1.330 (1.102–1.606)* |
| **HIV status** | - | - | |
| Known (Ref) | - | - | 1.00 |
| Unknown | - | - | 0.548 (0.426–0.704)* |
| **Pay for HIV services** | - | - | |
| Yes (Ref.) | - | - | 1.00 |
| No | - | - | 2.267 (1.470–3.496)* |

*Note*: *Significant at 95% (P<0.05).—not evaluated in that model, and *a*PRs = adjusted prevalence ratios.

prevention services. However, participants who did not know their HIV serostatus were 0.548 times less likely to use HIV prevention services than those who knew their HIV serostatus.

## Discussion

Curbing the HIV pandemic among transport workers partly requires sustainable utilization of multiple HIV services either singly or in combination [22]. Prior to this study, evidence about

the provision of HIV prevention services in combination by various stakeholders and the efficacy of using them existed. However, literature was scanty about the utilization of multiple HIV prevention services by transport workers. We found that less than half of the study participants (43%) utilized over one HIV prevention service in the past year preceding the study. Similarly, in a study among men who have sex with men (MSM) in Brazil, the utilization of behavioral and biomedical HIV prevention services was found to be as low as 2.8% [26]. In both contexts, the utilization were low, given that the success of HIV prevention services requires a high sustained uptake of these services.

Voluntary counselling and HIV testing is the first and entry point into other HIV prevention services such as ART, PREP, and EMTCT and these have been found to reduce risky sexual behaviors [5]. On the other hand, safe male circumcision must precede the use of condoms for effective HIV control. We found that most participants had tested for HIV and used condoms (16.2%), followed by those who had received safe male circumcision also used condoms (15%), and those who had tested for HIV and began on ART (9.1%). Figures reported in this study are still much lower than those reported by health facility-based studies conducted in Uganda among the general population. For example, a retrospective study conducted in eastern Uganda found that 92.4% of the participants who had tested and were diagnosed with HIV were immediately started on ART [23]. The low utilization of the multiple HIV prevention services in this study may be related to the COVID-19 pandemic restrictions that led to the suspension of community outreach services and camps, particularly for HIV testing (VCT), safe male circumcision (SMC), condom awareness and demand creation [2, 18]. In addition, the low integration of ARV-based prevention services such as PREP in many health facilities in Uganda may also be responsible for the low uptake of multiple HIV prevention services in this study [2].

Similarly, our analysis indicates that across single use, a relatively low proportion of the study participants used condoms (32.2%) followed by voluntary counselling and testing (27.1%), and safe male circumcision (SMC, 17.3%). However, the literature indicates that the utilization of HIV testing services among transport workers is much higher than the utilization of other HIV prevention services in other countries. In Zambia, the uptake of HIV counselling and testing (VCT) was reported at 83%, compared with 39% who used safe male circumcision (SMC), and 34.8 to 65% who used condoms in Nigeria [17, 35, 36]. Although previous studies conducted in Uganda among motorcycle taxi riders and truck drivers found the use of VCT and condoms to be as high as 87%, our study indicates a much lower figure, particularly in a city where the HIV rates are high, which is deeply concerning and requires shift actions to raise the demand for use of HIV prevention services [27, 37, 38].

Across demographic characteristics, we found that age, religion, marital status, number of concurrent sex partners, awareness of HIV status, and length of stay in the area were statistically significantly associated with the utilization of multiple HIV prevention services. This partly resonates with similar results in Zambia where multiple sex partners and marital status were found to be associated with the uptake of HIV testing services among long-distance truck drivers [35]. Whereas age, serostatus, stigma, and HIV knowledge were associated with the use of VCT services among motorcycle taxi riders in Uganda [27]. Additionally, knowledge of HIV prevention, perceived risk of HIV acquisition, and higher education levels were statistically significantly associated with the utilization of condoms among truck drivers in Togo and Nigeria [17, 28].

However, from our multivariate analysis only religion, the number of sex partners, awareness of HIV prevention services, awareness of one's serostatus, and paying for HIV services significantly predicted the use of multiple HIV prevention services. We found that being a Muslim or a person of another faith was associated with higher likelihood of using multiple

HIV prevention services compared with participants who subscribe to the Christian faith. Our possible explanation for this finding may lay in the fact that Uganda is a conservative community, with certain religious beliefs that prevent the use of certain specific HIV prevention services that act as artificial family planning methods, such as condoms among Catholics. Thus, strategies that aim to increase the use of multiple HIV services need to involve the religious community, most especially of the Christian faith. Similar to a study conducted in Zambia about condom use among long-distance truck drivers, we found that participants who had two or more concurrent sex partners compared with those who had single, or no partners had higher likelihood of using multiple HIV prevention services [35]. Those with multiple partners may understand the risky behaviors they are engaged in and take serious precautions to avoid acquiring HIV, but it may also indicate the effectiveness of the messages and current campaigns by various government and non-government players to combat the HIV scourge.

This study found that awareness of two or more HIV prevention services was associated with higher likelihood of using multiple HIV prevention services compared with the awareness of only one HIV prevention service. This finding calls for mention of available HIV prevention services each time health workers have an interface with the transport workers. However, because of the limited time transport workers have, their interface with health workers may be minimal. Thus, other platforms need to be explored such as through their associations, leadership, or social media. Using mainstream media (radio and television), other than other targeted approaches mentioned above, in creating a demand for HIV prevention services, may not be an effective strategy, particularly in mobile workers, such as transport workers, who always have limited time and access to such services at their workplaces.

Knowing one's HIV status has been found to reduce the risk of HIV, partly through reducing risky sexual behaviors. Also, it increases the awareness and utilization of HIV services, especially those linked to an HIV diagnosis such as PEP, PREP, ART, and Emtct [5]. Similarly, we found that participants who did not know their Serostatus were less likely to use multiple HIV prevention services than those who knew their HIV Serostatus, as earlier reported in Zambia, and Uganda among truck drivers and motorcycle taxi riders using HIV testing services [27, 35].

The findings of this study indicate that paying for HIV prevention services is a barrier whereas obtaining free services may be a facilitator. Specifically, we found that participants who did not pay to access the HIV prevention services were 2.267 times more likely to use multiple HIV prevention services than those who paid to access the HIV prevention services. Given the current tough economic times brought by the COVID-19 pandemic, stakeholders must make additional investments in the providing free HIV prevention services, if we are to overcome the current HIV epidemic in sub-Saharan Africa. However, the free services need to be provided innovatively targeting the transport workers given that they have limited time. For example, transport workers may access services such as condom use through their leadership, offices, commercial stages, and workshops. In summary, this study suggests that behavioral factors influence the use of multiple HIV prevention services compared with other individual factors.

## Strength and limitation of the study

This study sheds light on the use of multiple HIV prevention services across the various transport workers tracking long, short, and medium distances unlike the previous studies. Therefore, the findings of the study can be generalized to countries in sub-Saharan Africa with similar health care and transport systems (with the three classes of drivers) as Uganda. This study also has limitations that may influence the conclusions made in this paper. First, we also

did not assess the health system factors related to the use of HIV prevention services, which could have enriched the study. The cross-sectional study design does not also allow for causal attributions. Lastly, the administration of the questionnaire by interviewers carried the risk of social desirability bias.

## Conclusions

Our study conducted in Mbarara city-southwestern Uganda found suboptimal uptake of multiple HIV prevention services among transport workers. The study also found that factors that influence the use of the multiple HIV prevention services included religion, awareness of HIV prevention services, knowledge of HIV status, number of sex partners and experience of paying for HIV prevention services. Therefore, differentiated strategies are needed to increase the utilization of HIV prevention services among the transport workers to curb the HIV epidemic. We recommend interventions aimed at increasing awareness about HIV prevention services, awareness about HIV prevention service-related costs to be borne by the patients, encouraging HIV testing among persons with multiple sex partners and Christians. Future qualitative research to explore attitudes and perceptions regarding HIV prevention services is also recommended.

## Supporting information

**S1 Data. Analyzed data set in SPSS.**
(SAV)

## Acknowledgments

Special thanks to the transport workers and their leadership, the city clerk, and the Director of Health Services Mbarara district where the study was conducted.

## Author Contributions

**Conceptualization:** Benjamin Betunga, Phionah Atuhaire, Catherine Nakasiita, Christa Kanyamuneza, Proscovia Namiiro, Joseph Tugume, Matovu Hairat, John Baptist Asiimwe.

**Data curation:** Proscovia Namiiro.

**Formal analysis:** Benjamin Betunga, Phionah Atuhaire, Joseph Tugume, Matovu Hairat, Ahmed M. Sarki, Benedicto Mugabi, Birungi Lilian, Richard Mugisha, Edward Kumakech, John Baptist Asiimwe.

**Funding acquisition:** Benjamin Betunga, Phionah Atuhaire, Catherine Nakasiita, Christa Kanyamuneza, Proscovia Namiiro, Joseph Tugume, Matovu Hairat, John Baptist Asiimwe.

**Investigation:** Benjamin Betunga, Phionah Atuhaire, Catherine Nakasiita, Christa Kanyamuneza, John Baptist Asiimwe.

**Methodology:** Benjamin Betunga, Catherine Nakasiita, Christa Kanyamuneza, Proscovia Namiiro, Joseph Tugume, Matovu Hairat, Ahmed M. Sarki, Benedicto Mugabi, Birungi Lilian, Richard Mugisha, Edward Kumakech, John Baptist Asiimwe.

**Project administration:** Catherine Nakasiita, Christa Kanyamuneza, Proscovia Namiiro, Matovu Hairat.

**Supervision:** Ahmed M. Sarki, Richard Mugisha, Edward Kumakech, John Baptist Asiimwe.

**Writing – original draft:** Benjamin Betunga, Phionah Atuhaire, Catherine Nakasiita, Christa Kanyamuneza, Proscovia Namiiro, Joseph Tugume, Matovu Hairat, Ahmed M. Sarki, Benedicto Mugabi, Birungi Lilian, Richard Mugisha, Edward Kumakech, John Baptist Asiimwe.

**Writing – review & editing:** Benjamin Betunga, Phionah Atuhaire, Catherine Nakasiita, Christa Kanyamuneza, Proscovia Namiiro, Joseph Tugume, Matovu Hairat, Ahmed M. Sarki, Benedicto Mugabi, Birungi Lilian, Richard Mugisha, Edward Kumakech, John Baptist Asiimwe.

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
