## [Decision Letter · Decision Letter 0]

21 Dec 2022

PGPH-D-22-01790

Factors influencing the use of multiple HIV prevention services among Transport workers in a City in Southwestern Uganda.

Dear Dr. Asiimwe,

Thank you for submitting your manuscript to PLOS Global Public Health. After careful consideration, we feel that it has merit but does not fully meet PLOS Global Public Health’s publication criteria as it currently stands. Therefore, we invite you to submit a revised version of the manuscript that addresses the points raised during the review process.

We look forward to receiving your revised manuscript.

Kind regards,

Lei Gao

Academic Editor

Journal Requirements:

1. PLOS Global Public Health does not copy edit accepted manuscripts (https://journals.plos.org/globalpublichealth/s/criteria-for-publication#loc-5). To that effect, please ensure that your submission is free of typos and grammatical errors.

Additional Editor Comments (if provided):

Reviewers' comments:

Reviewer's Responses to Questions

**Comments to the Author**

1. Does this manuscript meet PLOS Global Public Health’s publication criteria? Is the manuscript technically sound, and do the data support the conclusions? The manuscript must describe methodologically and ethically rigorous research with conclusions that are appropriately drawn based on the data presented.

Reviewer #1: Yes

Reviewer #2: Yes

2. Has the statistical analysis been performed appropriately and rigorously?

Reviewer #1: Yes

Reviewer #2: Yes

3. Have the authors made all data underlying the findings in their manuscript fully available (please refer to the Data Availability Statement at the start of the manuscript PDF file)?

Reviewer #1: Yes

Reviewer #2: Yes

4. Is the manuscript presented in an intelligible fashion and written in standard English?

Reviewer #1: Yes

Reviewer #2: Yes

5. Review Comments to the Author

Reviewer #1: This is a very interesting and well written paper.

i have a few comments:

"Like in other sub-Saharan countries, the decline in the prevalence of 67 HIV is partly attributed to the use of HIV prevention services such as condom use, and 68 antiretroviral based prevention services."

=> Yes and many deaths which results in a smaller numerator… you should mention this

in this cross sectional analysis odds ratios are used perhaps prevalence ratios would be more appropriate (the significant results would not change) using a modified poisson. ORs overestimated the RR in such studies.

"...the findings show that most of the participants 340 had tested for HIV and then used condoms (16.2%),"

=>most and 16.2% seem to contradict each other 1 in 6!

discussion “Nevertheless, our analysis indicates that across single use, most study participants used condoms 474 (32.2%) followed by voluntary counseling and testing (27.1%), and safe male circumcision (SMC, 475 17.3%). »

=>again most and 27.1% i am struck about how low it is. please rephrase it hides the fact that it is very low!

perhaps the discussion focusses on those who used different methods but perhaps emphasize the low use of any single method which is an important message too.

perhaps trying to identify subgroups doing a PCA or MCA analysis would lead to insights

how many dimensions drive variability (screeplot and the kaiser rule) what variables load on these dimensions and what does this capture... it may add to your study

Reviewer #2: The manuscript is well written.

There is the logical flow of ideas, and it is reader friendly.

Keeps the reader glued to the end.

You have provided a detailed description of all the methodological process to help other authors replicate the study.

The discussion is well done. You gave a cautious overall interpretation of results considering objectives, limitations, multiplicity of analyses, results from similar studies, and other relevant evidence

You have also discussed the limitations of the study, taking into account sources of potential bias.

The conclusion is well written and reflect the major findings of the study.

Congratulations and thank you for the extra effort put into this paper.

i have just two comments for you to address based on the recommendations of STROBE.

1) Please clearly emphasize the strength of this study.

2) Please discuss the generalisability of the study results.

6. PLOS authors have the option to publish the peer review history of their article (what does this mean?). If published, this will include your full peer review and any attached files.

**Do you want your identity to be public for this peer review?** For information about this choice, including consent withdrawal, please see our Privacy Policy.

Reviewer #1: No

Reviewer #2: **Yes: **Dorothy Boakye

---

## [Decision Letter · Decision Letter 1]

9 Feb 2023

Factors influencing the use of multiple HIV prevention services among Transport workers in a City in Southwestern Uganda.

PGPH-D-22-01790R1

Dear Mr., Asiimwe,

We are pleased to inform you that your manuscript 'Factors influencing the use of multiple HIV prevention services among Transport workers in a City in Southwestern Uganda.' has been provisionally accepted for publication in PLOS Global Public Health.

Best regards,

Lei Gao

Academic Editor

Reviewer Comments (if any, and for reference):

Reviewer's Responses to Questions

**Comments to the Author**

1. If the authors have adequately addressed your comments raised in a previous round of review and you feel that this manuscript is now acceptable for publication, you may indicate that here to bypass the “Comments to the Author” section, enter your conflict of interest statement in the “Confidential to Editor” section, and submit your "Accept" recommendation.

Reviewer #1: All comments have been addressed

Reviewer #2: All comments have been addressed

2. Does this manuscript meet PLOS Global Public Health’s publication criteria? Is the manuscript technically sound, and do the data support the conclusions? The manuscript must describe methodologically and ethically rigorous research with conclusions that are appropriately drawn based on the data presented.

Reviewer #1: Yes

Reviewer #2: Yes

3. Has the statistical analysis been performed appropriately and rigorously?

Reviewer #1: Yes

Reviewer #2: Yes

4. Have the authors made all data underlying the findings in their manuscript fully available (please refer to the Data Availability Statement at the start of the manuscript PDF file)?

Reviewer #1: Yes

Reviewer #2: Yes

5. Is the manuscript presented in an intelligible fashion and written in standard English?

Reviewer #1: Yes

Reviewer #2: Yes

6. Review Comments to the Author

Reviewer #1: (No Response)

Reviewer #2: congratulations to the authors as they embark on more exciting research on HIV in Uganda and Beyond

7. PLOS authors have the option to publish the peer review history of their article (what does this mean?). If published, this will include your full peer review and any attached files.

**Do you want your identity to be public for this peer review?** For information about this choice, including consent withdrawal, please see our Privacy Policy.

Reviewer #1: No

Reviewer #2: No
